# Evaluation of leukopenia during sepsis as a marker of sepsis-defining organ dysfunction

**Samuel H. Belok** [ID]*, **Nicholas A. Bosch** [ID]**, Elizabeth S. Klings, Allan J. Walkey**

Pulmonary Center, Boston University School of Medicine, Boston, Massachusetts, United States of America

* samuel.belok@bmc.org

## Abstract

### Background

Although both leukocytosis and leukopenia have been considered Systemic Inflammatory Response Syndrome criteria, leukopenia is not generally considered a normal response to infection. We sought to evaluate the prognostic validity of leukopenia as a sign of sepsis-defining hematological organ dysfunction within the Sepsis-3 framework. We hypothesized that leukopenia is associated with higher risk of mortality than leukocytosis among patients with suspected infection.

### Methods

We performed a retrospective cohort study using the Medical Information Mart v1.4 in Intensive Care-III database. Multivariable regression models were used to evaluate the association between leukopenia and mortality in patients with suspected infection defined by Sepsis-3.

### Results

We identified 5,909 ICU patients with suspected infection; 250 (4.2%) had leukopenia. Leukopenia was associated with increased in-hospital mortality compared with leukocytosis (OR, 1.5; 95% CI 1.1–1.9). After adjusting for demographics and comorbidities in the Sepsis-3 consensus model, leukopenia remained associated with increased risk of mortality compared with leukocytosis (OR 1.6, 95% CI 1.2–2.2). Further adjustment for the platelet component of the SOFA attenuated the association between leukopenia and mortality (OR decreased from 1.5 to 1.1). However, 83 (1.4%) of patients had leukopenia without thrombocytopenia and 14 had leukopenia prior to thrombocytopenia.

### Conclusions

Among ICU patients with suspected infection, leukopenia was associated with increased risk of death compared with leukocytosis. Due to correlation with thrombocytopenia, leukopenia did not independently improve the prognostic validity of SOFA; however, leukopenia may present as a sign of sepsis prior to thrombocytopenia in a small subset of patients.

**Data Availability Statement:** The MIMIC III ('Medical Information Mart for Intensive Care') database is a freely accessible database containing individual deidentified patient data for patients admitted to critical care units between 2001 and

2012 at a single academic center (Beth Israel Deaconess Medical Center). Additional information about the MIMICIII dataset can be found here: https://www.nature.com/articles/sdata201635?source=post_page. Permission to use the dataset was granted to the authors through the PhyioNet Credential Health Data Use Agreement 1.5.0 for the MIMIC-III Clinical Database (v1.4) on December 18th, 2019. The necessary steps and contact information to apply to gain access to the data are found here: https://mimic.mit.edu/iii/gettingstarted/.

**Funding:** The authors received no specific funding for this work.

**Competing interests:** The authors have declared that no competing interests exist.

# Background

Prior to 2016, [1] consensus definitions conceptualized sepsis as suspected infection with evidence of systemic inflammatory response syndrome (SIRS) [2]. White blood cell counts were an essential component of SIRS, with both leukocytosis and leukopenia being classified as a sign of SIRS. With increasing awareness that SIRS did not identify patients with a dysfunctional, life threatening response to infection [3–5], the Third International Consensus Definitions for Sepsis and Septic Shock (Sepsis-3) removed SIRS as a criteria used to define sepsis. In place of SIRS, measures of acute organ dysfunction with strong prognostic validity for life-threatening conditions were chosen to define sepsis [4].

Although studies from the 1990s [5–7] suggested that leukopenia during infection may be associated with poor outcomes, the incidence and prognostic validity of leukopenia within a contemporary cohort and the Sepsis-3 conceptual framework is unclear. Identifying leukopenia as an acute organ dysfunction, rather than as a SIRS criteria, could have ramifications for early identification and expedited treatment of patients with time-sensitive sepsis. We sought to evaluate the association between leukopenia and mortality among critically ill patients with suspected infection to test the hypothesis that leukopenia acts as a sign of organ dysfunction—rather than SIRS—within the Sepsis-3 conceptual framework.

# Materials and methods

## Cohort

We performed a retrospective cohort study of Intensive Care Unit (ICU) admissions for suspected infection between 2008 and 2012 using the Medical Information Mart in Intensive Care (MIMIC)-III database v1.4 [8–10]. Patients were defined as having suspected infection upon ICU admission as per Sepsis-3 criteria—if antibiotics were prescribed and body fluid culture was obtained within a 72 hour time frame. "Onset" of suspected infection was defined as the time at which the first of these two events occurred. [4] Patients with suspected infection, rather than a diagnosis of sepsis, were evaluated in order to explore the added clinical utility and prognostic validity of leukopenia as a sepsis-defining organ dysfunction—that is, identifying patients with "sepsis" who may not have been identified with the traditional Sepsis-3 definition.

We excluded patients with comorbidities that may lead to chronic leukopenia that may confound the evaluation of acute leukopenia as a sepsis-defining organ dysfunction (e.g., HIV disease, hematologic malignancy solid organ malignancy and/or metastases, post-organ transplant, alcohol use, hepatic dysfunction, or receipt of bone marrow stimulating agents [filgastrim or sargramostim]) (Fig 1). Additionally, patients who developed suspected infection more than 24 hours after admission to the ICU and those with a minimum WBC <1000/μL were excluded in order to minimize confounding by other processes that could contribute to leukopenia.

## Exposures, covariates, and outcomes of interest

We determined exposures and covariates in the time window from 48 hours before to 24 hours after the onset of suspected infection. We determined and categorized minimum WBC count based on the reference laboratory values as follows: Leukopenia (WBC <4000/μL), normal (WBC 4000–10,000 /μL), and leukocytosis (WBC >10,000 /μL). In sensitivity analyses, WBC counts were modeled as continuous variables using splines.

We identified covariates analogous to those used in the Sepsis-3 study baseline model: [4] (age [fractional polynomial], race [Black, White, Hispanic, Asian, other, unknown],

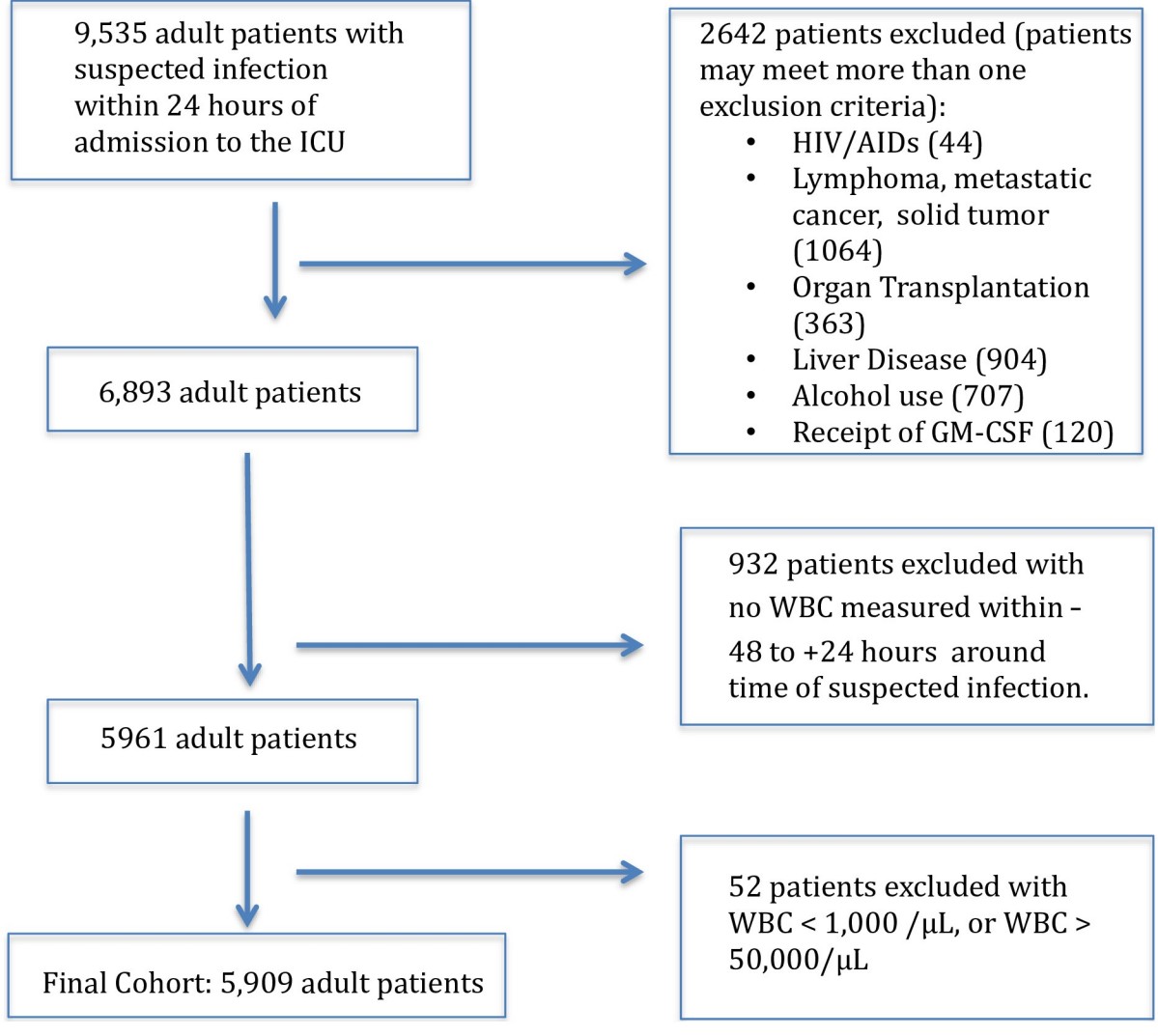

**Fig 1. Eligibility and exclusion criteria consort diagram.**

Elixhauser-van Walraven comorbidity score [fractional polynomial] [11], gender). The maximum SOFA score was calculated between 48 hours prior to suspected infection until 24 hours post-infection. In-hospital mortality was the primary outcome of the study.

## Statistical methods

Baseline characteristics and values were reported as means +/- standard deviations or percentages, as appropriate. We fitted natural splines to visualize the relationship between mortality rate and WBC count. Nested logistic regression models were used to determine changes to the association between WBC (with leukocytosis as the reference group) and mortality as additional potential confounding variables were added to the analysis and to reflect the processes of model development for the Sepsis-3 definition [4]. We analyzed 1) a univariable model with WBC category alone; 2) a multivariable model that included WBC category and covariates from the Sepsis-3 baseline model; 3) a multivariable model that included WBC category, Sepsis-3 baseline model components, and the SOFA score; and 4) a model including WBC

category and SOFA score. For each model, we reported the odds ratio for hospital mortality associated with leukopenia compared to leukocytosis.

### Exploratory analysis

We used Spearman's ranl correlation to explore the relationship between existing SOFA components and WBC count. We then constructed models for hospital mortality adding each individual organ component of the SOFA score to WBC category to quantify the change in the beta-coefficient for leukopenia attributed to each SOFA component. Additionally, we repeated our analyses by substituting neutropenia (defined as neutrophil count less than 1700/μL) and lymphopenia (defined as lymphocyte count less than 900/μL) for leukopenia among patients with a complete blood count differential, in order to assess if the association between WBC and mortality are driven by specific WBC components. We assessed correlation between neutrophil count and total WBCs and between lymphocyte count and total WBCs using Pearson's correlation based on the linearity of the association.

### Sensitivity analysis

We conducted a sensitivity analysis where we restricted our cohort to patients with an ICD-9 code for infection, to increase the likelihood that the cohort included patients with confirmed, rather than suspected, infection.

All analyses were performed with R-studio Version 1.1.456 (RStudio, Inc., Boston, MA), two-sided alpha level 0.05. The study was considered non-human subjects research by the Boston University Institutional Review Board.

### Ethics statement

This study was designated by the Boston University IRB as not Human Subjects Research. All data was fully anonymized prior to us accessing the data.

## Results

### Patient characteristics (Table 1)

We identified 5,909 ICU patients with suspected infection. The average age of the cohort was 68.3±16.7 years old, with a mean SOFA score of 8.5±3.69. 3027 patients (51.2%) had leukocytosis and 250 (4.2%) had leukopenia. 1,081 (18.3%) patients died in the hospital. Mortality was 27.2% for patients with leukopenia, 20.4% for patients with leukocytosis and 15.0% for patients with normal WBC.

### Outcomes associated with leukopenia (Table 2)

The association between WBC and mortality was U-shaped (Fig 2); the lowest mortality rates occurred in patients with a WBC count approximating 5000/μL. In unadjusted analysis, leukopenia was associated with increased in-hospital mortality compared with leukocytosis (odds ratio [OR], 1.5; 95% CI 1.1–1.9). After adjusting for covariates from the Sepsis-3 baseline model (i.e., age, race, gender and comorbidities), leukopenia remained associated with increased risk of mortality compared with leukocytosis (OR 1.6, 95% CI 1.2–2.2). After adjusting for the SOFA score, the association between leukopenia and mortality was attenuated (OR, 1.1; 95% CI 0.8–1.5) (Table 2). In all models, normal WBC was associated with lower mortality compared to leukocytosis (OR 0.7, 95% CI 0.6–0.8).

**Table 1. Characteristics of patients admitted to the ICU for suspected infection between 2008 and 2012 (N = 5,909).**

| | |
|---|---|
| Age (mean (SD)) | 68.3 (16.7) |
| Gender Female (%) | 2940 (49.8%) |
| Race/Ethnicity | |
| Black | 593 (10.0%) |
| White | 4346 (73.5%) |
| Hispanic | 172 (2.9%) |
| Asian | 140 (2.4%) |
| Other/Unknown | 658 (11.2%) |
| Elixhauser Comorbidity Index (mean (SD)) | 7.8 (6.63) |
| Maximum SOFA score (mean (SD)) | 8.5 (3.69) |
| Minimum WBC (mean (SD)) | 12.5 (6.72) |
| Maximum WBC (mean (SD)) | 14.7 (8.21) |
| WBC Category | |
| Leukocytosis | 3027 (51.2%) |
| Median WBC (IQR) | 15.2 (1280–1940 /μL) |
| Normal | 2632 (44.5%) |
| Median WBC (IQR) | 8200 (6500–9600 /μL) |
| Leukopenia | 250 (4.2%) |
| Median WBC (IQR) | 2700 /μL (2000-3400/μL) |
| In-hospital Mortality—entire cohort (%): | 1081 (18.3%) |
| Leukocytosis | 316 (20.4%) |
| Normal WBC | 394 (15.0%) |
| Leukopenia | 68 (27.2%) |
| Infection type: (by ICD codes) | |
| Culture positive bacteremia | 639 (10.8%) |
| Gangrene | 24 (0.4%) |
| Infected Device | 365 (6.2%) |
| Intra-abdominal infection | 333 (5.6%) |
| Invasive fungal disease | 47 (0.8%) |
| Obstructive Lung Disease Exacerbation | 205 (3.5%) |
| Other | 42 (0.7%) |
| Pericarditis/endocarditis | 68 (1.2%) |
| Phlebitis | 23 (0.4%) |
| Pneumonia/Pneumococcus | 1542 (26.1%) |
| Post-operative infection | 138 (2.3%) |
| Pyelonephritis/genitourinary | 793 (13.4%) |
| Skin/joint/soft tissue infection | 150 (2.5%) |
| Unspecified septicemia, bacteremia | 1446 (24.5%) |
| Upper respiratory tract infection | 30 (0.5%) |
| N/A | 5 (0.01%) |

*Definition of abbreviation*: SOFA = sequential organ failure assessment; SD = standard deviation; WBC = white blood cell count.

## Exploratory analysis: Sequential analysis of individual organ dysfunction scores, leukopenia, and mortality

The coagulation component of the SOFA score (platelet count) was most responsible for attenuating the association between leukopenia and mortality (Table 3). Platelet count and WBC

**Table 2. Odds ratios for the logistic regression models.**

| Covariate | OR (C.I.) | p-value |
|---|---|---|
| Leukopenia (model 1 -unadjusted) | 1.5 (1.1–1.9) | 0.01 |
| Normal WBC | 0.7 (0.6–0.8) | <0.001 |
| Leukocytosis *(reference)* | 1 | - |
| Leukopenia (model 2—Sepsis-3 baseline model covariates) | 1.6 (1.2–2.2) | 0.002 |
| Normal WBC | 0.7 (0.6–0.8) | <0.001 |
| Leukocytosis *(reference)* | 1 | - |
| Leukopenia (model 3—Sepsis-3 baseline model and maximum SOFA score covariates) | 1.1 (0.8–1.5) | 0.54 |
| Normal WBC | 0.7 (0.6–0.8) | <0.001 |
| Leukocytosis *(reference)* | 1 | - |
| Leukopenia (model 4 –maximum SOFA score covariates) | 1.0 (0.7–1.3) | 0.9 |
| Normal WBC | 0.7 (0.6–0.8) | <0.001 |
| Leukocytosis *(reference)* | 1 | - |

Covariates included in each model included.

Model 1: WBC category.

Model 2: WBC category, age (fractional polynomial), race, gender, co-morbidity index (fractional polynomial).

Model 3: WBC category, age(fractional polynomial), race, gender, SOFA score, co-morbidity index (fractional polynomial).

Model 4: WBC category, SOFA score.

*Definition of abbreviation*: SOFA = sequential organ function assessement; SD = standard deviation; WBC = white blood cell count; OR = Odds Ratio; CI = Confidence Interval.

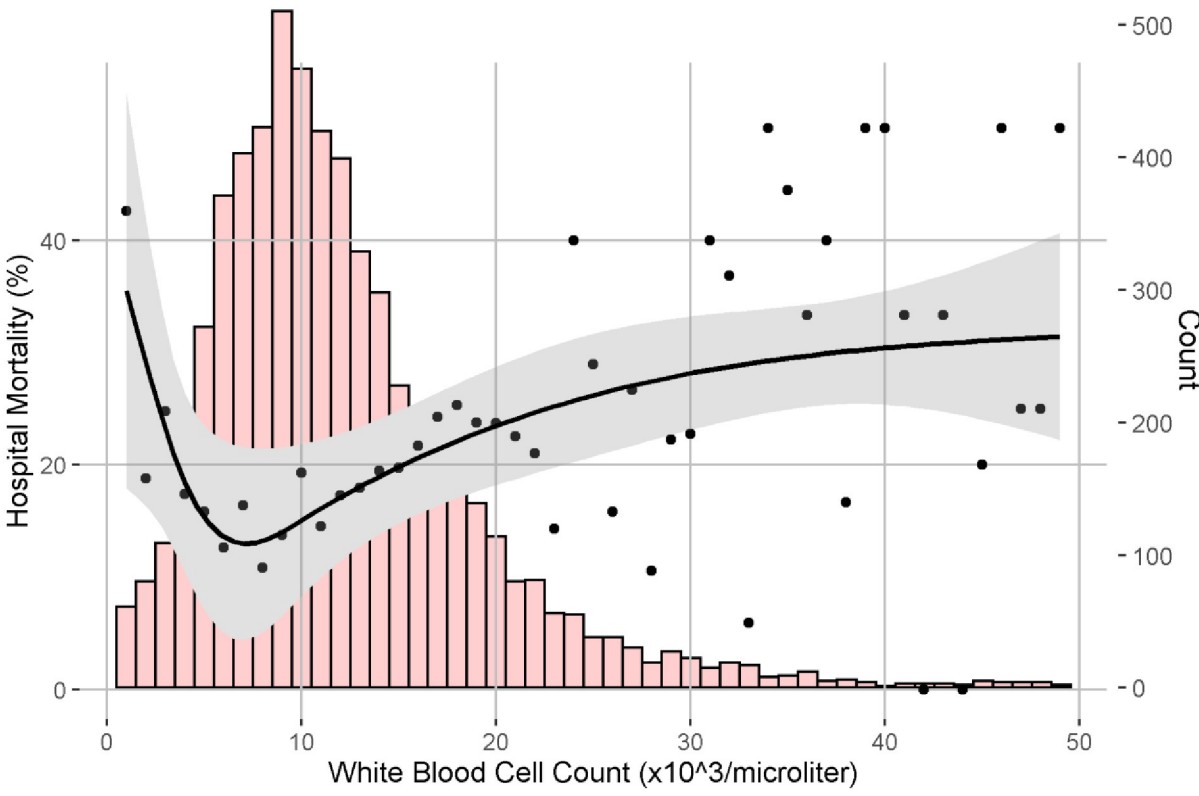

**Fig 2. Mortality of patients with suspected infection plotted in the solid black line with 95% confidence intervals denoted by the grey.** The mortality plot overlays a histogram of patients admitted to the ICU with suspected infection denoted by the red bar graphs and correspond to the Y-axis on the right side.

**Table 3. Beta-estimates before and after adjustment for specific organ dysfunction.**

| Beta-Estimate for leukopenia's effect on mortality prior to adjustment | Beta-Estimate for leukopenia's effect on mortality following adjustment for listed organ dysfunction | Beta estimate change |
|---|---|---|
| 0.37 | 0.03 (coagulation) | -92% change |
| 0.37 | 0.28 (respiratory) | -24% change |
| 0.37 | 0.23 (cardiovascular) | -38% change |
| 0.37 | 0.30 (liver) | -19% change |
| 0.37 | 0.32 (neurologic) | -14% change |
| 0.37 | 0.39 (renal) | +5% change |

were positively correlated (Spearman's Rank Correlation Rho = 0.31, p <0.001) (Fig 3). Among patients with suspected infection and leukopenia, 83 (1.4%) had leukopenia without thrombocytopenia, 167 (2.8%) had leukopenia with thrombocytopenia with 14 (0.2%) having leukopenia prior to thrombocytopenia.

## Exploratory analysis: Neutropenia, leukopenia and mortality

As an additional exploratory analysis, we evaluated the presence of neutropenia and lymphopenia—rather than leukopenia—among 5310 (90%) patients with complete blood count differentials. Of these patients, 167 (3.1%) had neutropenia (<1700 /μL), 1563 had a normal neutrophil levels (1700 /μL– 7000 /μL), and 3580 (67.4%) had neutrocytosis (>7000 /μL). The median neutrophil count was 9000 /μL (IQR 6000–1310). In addition, 2642 (49.8%) patients had lymphopenia (< 900 /μL), 2393 (45.1%) had a normal lymphocyte level (900–1700 /μL), and 275 (5.1%) patients had a lymphocytosis >1700 /μL. The median lymphocyte count was 900 /μL (IQR 500–1500).

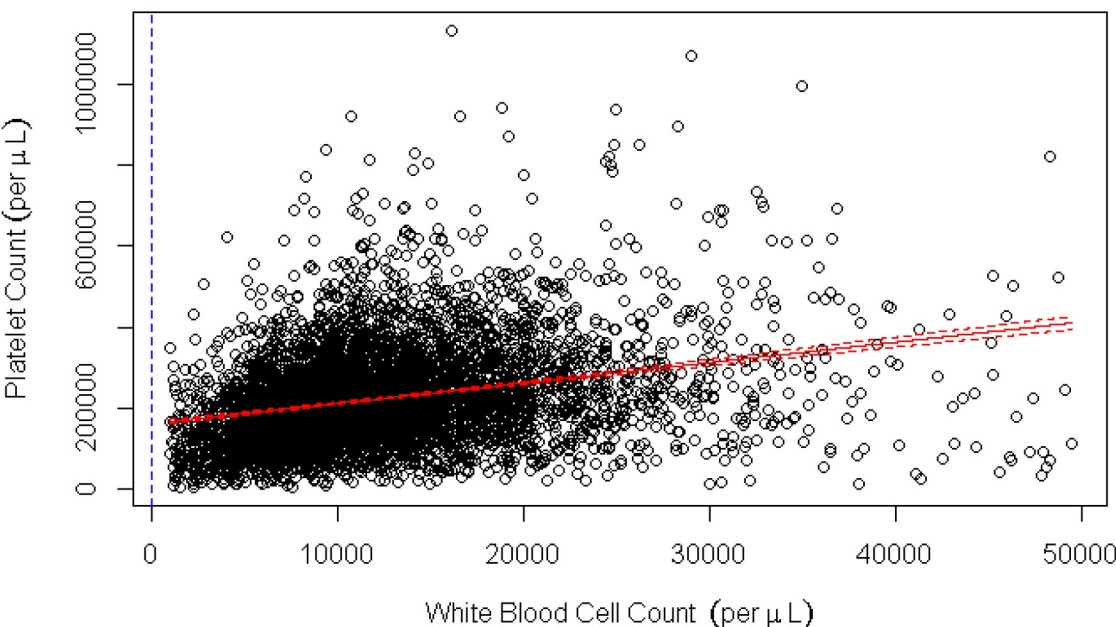

**Fig 3. Correlation of platelet count with white blood cell count.** Spearman's Rank Correlation Rho = 0.313.

In the unadjusted analysis, neutropenia was associated with increased in-hospital mortality compared with neutrocytosis (OR 1.9; 95% CI 1.4–2.7). After adjusting for covariates from the Sepsis-3 baseline model (i.e., age, race, gender and comorbities), leukopenia remained associated with increased risk of mortality compared with leukocytosis (OR 2.3, 95% CI 1.6–3.2). After further adjusting for the SOFA score covariates neutropenia remained associated with increased risk of mortality (OR 1.6 95% CI 1.1–2.3) (S1 Table). Lymphopenia, on the other hand, was not associated with increased in-hospital mortality in any of the unadjusted or adjusted models (S2 Table). Pearson's correlation coefficient between total WBC count and neutrophils was 0.92 (95% CI 0.92–0.93, p = <0.001) and between WBC and lymphocytes was 0.25 (95% CI 0.22–0.27, p = <0.001).

## Sensitivity analysis

Five of 5,909 admitted to the ICU with suspected infection did not have ICD-9 codes associated with a potential infectious source; results were unchanged excluding these patients (n = 5904).

## Discussion

In order to evaluate the potential role of leukopenia as a marker of life threating organ dysfunction within the conceptual model of Sepsis-3, we determined the association between leukopenia and mortality among patients admitted to the ICU with suspected infection. Leukopenia was rare, but associated with increased risk of death as compared with leukocytosis in unadjusted analyses. However, associations between leukopenia and mortality were strongly attenuated by the platelet component of the SOFA score. Because leukopenia has similar prognostic validity to thrombocytopenia during suspected infection, and is present in some patients prior to onset of thrombocytopenia, leukopenia is potentially a clinically important marker of life threatening hematological dysfunction in patients with suspected infection.

Few studies have investigated the relationship between mortality and leukopenia. Georges et al. demonstrated that in patients with community acquired pneumonia, there was an increased mortality risk in patients with WBC <4000/ μL. Similarly, Leibovici et al. demonstrated increased mortality in patients with bacteremia who had granulocyte counts <1000 / μL compared to those with granulocyte counts between 1000-4000/μL or 4000-8000/μL, [6]. Additionally, Knaus et al. found that in leukopenia was present among more non-survivors (15%) than survivors (7%) with sepsis [7]. Our findings add to prior studies by demonstrating increased mortality in ICU patients with suspected infection and leukopenia (and more importantly neutropenia) compared to ICU patients with suspected infection and leukocytosis.

Cytopenias during sepsis may result from decreased bone marrow production or increased destruction [12]. In the bone marrow, both leukopenia and thrombocytopenia may arise due to maturation arrest, inadequate bone marrow supply of progenitors [6], or hemophagocytosis [13]. In the peripheral blood, leukocyte-platelet interaction during sepsis [14] may similarly cause leuko- and thrombocytopenias. Platelets and neutrophils form neutrophil-extracellular traps (NETs) for bacteria [15] which may lead to both active leukocyte destruction [16] and thrombocytopenia [17]. Whether our novel findings of correlation and similar prognostic validity between thrombocytopenia and leukopenia during sepsis represent similar levels of bone marrow dysfunction, drug toxicity, sequestration, or consumption requires further study. However, results from our exploratory analysis showed that neutropenia—but not lymphopenia—was strongly associated with mortality after adjusting for the SOFA scores, suggesting that decreased leukocyte production is a less likely mechanism of the association between leukopenia and mortality during sepsis.

Our study has limitations. Using ICD-9 codes to exclude patients with etiologies of leukopenia other than sepsis may have imperfect sensitivity. Additionally, our data set only includes data on the day of arrival to ICU and after ICU admission, without the ability to evaluate laboratory values prior to ICU admission that may aid in prognostic evaluation.

## Conclusions

Among critically ill patients with suspected infection, leukopenia was rare, but associated with increased risk of death as compared with leukocytosis. However, the "coagulation" component of the SOFA score (as represented by the platelet count) accounted for the increased mortality risk associated with leukopenia, the presence of leukopenia did not appear to add prognostic information to the current Sepsis-3 criteria. Only a small subset of patients (<2%) would be captured on a modified SOFA score that includes a leukopenia category. Although leukopenia did not independently add to prognostic validity of the SOFA score above platelet count, the correlation of leukopenia with thrombocytopenia and association of neutropenia with mortality suggest that future studies should whether leukopenia or neutropenia may prospectively identify a larger group of patients earlier than current Sepsis-3 definitions.

## Supporting information

**S1 Table. Odds ratios for neutropenia models.**
(DOCX)

**S2 Table. Odds ratios for Lymphopenia models.**
(DOCX)

## Acknowledgments

### Declarations

We would like to acknowledge the work of researchers at the MIT Laboratory for Computational Physiology, collaborating research groups, patients and funders for making MIMIC a publicly available research resource.

## Author Contributions

**Conceptualization:** Samuel H. Belok, Nicholas A. Bosch, Elizabeth S. Klings, Allan J. Walkey.

**Data curation:** Samuel H. Belok, Nicholas A. Bosch, Allan J. Walkey.

**Formal analysis:** Samuel H. Belok, Nicholas A. Bosch, Allan J. Walkey.

**Investigation:** Samuel H. Belok, Nicholas A. Bosch, Allan J. Walkey.

**Methodology:** Samuel H. Belok, Nicholas A. Bosch, Elizabeth S. Klings, Allan J. Walkey.

**Project administration:** Nicholas A. Bosch, Allan J. Walkey.

**Resources:** Nicholas A. Bosch, Allan J. Walkey.

**Software:** Nicholas A. Bosch, Allan J. Walkey.

**Supervision:** Allan J. Walkey.

**Validation:** Samuel H. Belok, Nicholas A. Bosch, Allan J. Walkey.

**Visualization:** Samuel H. Belok, Allan J. Walkey.

**Writing – original draft:** Samuel H. Belok.

**Writing – review & editing:** Samuel H. Belok, Nicholas A. Bosch, Elizabeth S. Klings, Allan J. Walkey.

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
