## [Decision Letter · Decision Letter 0]

21 Dec 2020

PONE-D-20-27470

Evaluation of Leukopenia During Sepsis as a Marker of Sepsis-Defining Organ Dysfunction

PLOS ONE

Dear Dr. Belok,

Thank you for submitting your manuscript to PLOS ONE. After careful consideration, we feel that it has merit but does not fully meet PLOS ONE’s publication criteria as it currently stands. Therefore, we invite you to submit a revised version of the manuscript that addresses the points raised during the review process.

The reviewers have noted several limitations and asked for clarifications of data and analyses, as well as conclusions made in the manuscript.  The editor recommends a thoughtful addressing of each of the points raised in a MAJOR REVISION of the manuscript.

We look forward to receiving your revised manuscript.

Kind regards,

Scott Brakenridge, M.D.

Academic Editor

PLOS ONE

Journal Requirements:

1. Please ensure that your manuscript meets PLOS ONE’s style requirements, including those for file naming. The PLOS ONE style templates can be found at

2. Please include the date(s) on which you accessed the databases or records to obtain the data used in your study.

3. In your ethics statement in the Methods section and in the online submission form, please provide additional information about the data used in your retrospective study. Specifically, please ensure that you have discussed whether all data were fully anonymized before you accessed them and/or whether the IRB or ethics committee waived the requirement for informed consent. If patients provided informed written consent to have data from their medical records used in research, please include this information.

Reviewers' comments:

Reviewer's Responses to Questions

**Comments to the Author**

1. Is the manuscript technically sound, and do the data support the conclusions?

Reviewer #1: Partly

Reviewer #2: Yes

2. Has the statistical analysis been performed appropriately and rigorously? 

Reviewer #1: Yes

Reviewer #2: Yes

3. Have the authors made all data underlying the findings in their manuscript fully available?

Reviewer #1: No

Reviewer #2: No

4. Is the manuscript presented in an intelligible fashion and written in standard English?

Reviewer #1: Yes

Reviewer #2: Yes

5. Review Comments to the Author

Reviewer #1: Aim of this retrospective study in the ICU setting is to try to include leukopenia (WBCs< 4000 / uL , no mention of neutropenia) among the organ dysfunctions in the SEPSIS 3 definition. . “Patients with suspected infection,rather than a diagnosis of sepsis, were evaluated in order to explore the added clinical utility and prognostic validity of leukopenia as a sepsis-defining organ dysfunction – that is, identifying patients with “sepsis” who may not have been identified with the traditional Sepsis-3 definition” .LP per se has long been associated with poor outcome in case of infection. According to the AAs “ identifying leukopenia as an acute organ dysfunction, (rather than as a SIRS criteria), could have ramifications for early identification and expedited treatment of patients with time-sensitive sepsis”.

Leukopenia (LP) ”, the absolute reduced number of white blood cells in the peripheral blood: LP is almost always due to neutropenia or lymphopenia.The terms granulocytopenia and neutropenia are often used interchangeably. The definition of “leukopenia” varies, but in most laboratories the lower limit of a normal total white cell count is 3000/μl to 4000/μl. Neutropenia is defined as an absolute neutrophil count (ANC) of less than 1500/μl, an absolute count less than 800/μl being considered to be associated with an immunocompromised condition. New or severe leukopenia, especially neutropenia, mandates in any case an aggressive management , including a a thorough investigation to determine the cause / source of the leukopenia , to rule out infection / sepsis and an aggressive management (broad short term antiinfective if wise)

Please consider these notes I dare propose to the study

1. LP is since long a marker of severity and a well known risk factor for mortality and morbidity, particularly when severe and in case of absolute neutropenia < 800/ uL. I have not clear the advantage / amelioration of considering LP aa a marker of organ dysfunction (Bone Marrow[BM], I suppose), due to the above reasons . It should be considered per se!

2. The number of pts with LP is in general, and in this series in particular (average ICU pts), small : over 5,909 ICU patients with suspected infection, 250 (4.2%) had leukopenia . It is not specified a WBCs count “stratification” ( < 2000 / < 1000) and how many had severe LP: please, could you provide this figure?

3. SOFA median score was in this series 8.5 (mortality 15-20%). WBCs, not present in SOFA, are considered in SAPSII and APACHE II , and are relevant in case of count > 20000/ uL. Close to SOFA 8.5 score are SAPSII 38, , with a mortality of 21.35 ( checked with WBCs 3000 and 1000) and APACHE2 14 (mortality 15%, checked with WBCs count 3000 and 1000: source MEDCALC). According to the mortality prediction curves , minimal seems to be the impact of LP on the score and on mortality

4. Among the SOFA components “platelet count ” (thrombocytopenia. TCP) ) represent “dysfunctioning coagulation”, not a dysfunctioning BM, and, as reported by the AAS, “further adjustment for the platelet component of the SOFA attenuated the association between leukopenia and mortality (OR decreased from 1.5 to 1.1)”. (see page 7).

a. Indeed, the relevance of LP might be stressed (but the absolute WBCs count is important into my opinion and to be reported

b. according to the AAS, “ 83 (1.4%) of patients had LP without thrombocytopenia and 14 had LP prior to thrombocytopenia”: then LP should be considered an important early warning per se , whether or not considered as the result of a dysfunctioning organ (BM) , and as an early warning and with sepsis / infection suspicion, to be accordingly managed (optimal clinical practice, with further , deeper efforts aiming at a differential diagnosis with appropriate investigations and appropriate antiinfective therapy if wise ) .

5. The AAs reported a correlation between LP and TCP , able to reduce the prognostic validity of SOFA (OR 1.5 to 1.1); LP , possibly representing an earlier sign of infection / sepsis if compared to TCP, as above quoted, deserves indeed attention, not only in this small subset ofLP patients, but in general (personal opinion, but always applied in clinical practice )

6. The impact of LP on outcome, as already underlined, might be relevant : much more relevant, however, should be the absolute WBCs count (below 4000, vs below 3000 vs below 1000): one size could not fit all, and the different absolute count might have a different impact on outcome; stratification of WBCs is, into my opinion, mandatory

7. A way to reinforce the relevance of LP could be the presence of neutropenia instead of “LP” (more specific perhaps in the ICU pts (see Quentin G, Azoulay et al. Influence of neutropenia on mortality of critically ill cancer patients: results of a meta-analysis on individual data , Crit Care, 2018; 22: 326. 2018) : neutropenia ( neutrophil< 1000 u/L) was associated with poor outcome in cancer pts.

a. I dare suggest the AAS to consider in their study the level of severity of LP and , instead of LP or together with LP , neutropenia: perhaps it might have a greater impact on outcome, particularly in case of severe neutropenia (< 1000/uL or < 500)

Statistical Methods

Page 5 – A comment / explanation of the statistical methods could be wise, since I think this could be key of the above proposed aim(s) of this study and might shed light on it .

Results

Page 6 - We identified 5,909 ICU patients with suspected infection. The average age of the cohortwas 68.3±16.7 years old, with a mean SOFA score of 8.5±3.69. 3027 patients (51.2%) had leukocytosis and 250 (4.2%) had leukopenia. 1,081 (18.3%) patients died in the hospital1,081 (18.3%) patients died in the hospital

I dare pose two questions :

How many of the 250 leukopenic pts died in Hospital?

How many leukopenic pts were not identified by Sepsis3 Criteria and died because of sepsis ?

Any difference in mortality between leukopenic pts with absolute WBCs count below 3000 u/L and below 1000 u/L?

Discussion

Page 7 –

• if LP is strongly attenuated by TCP, what is the relevance of its use in the score? The presence of LP has to be taken in any case into account as an early, relevant warning : perhaps the absolute number might impact more than the “absolute” definition . I dare ask a comment

Page 8 -

• Dealing with granulocytopenia (Leibovici et al ,1995), see also Quentin et al, 2018, above quoted , for a short comment : any idea of granulocytopenia and its impact on mortality in this series ?

• I totally agree about the BM dysfunction as a reason for TCP (any reason, as appropriately pointed out by the AAS in the draft), which could share the same cause of LP : but in SOFA the TCP is “dysfunctioning coagulation” (perhaps more appropriate “defective haemostasis”!) and not BM.

Page 9 -

Conclusion

• I am sorry to say I am not convinced at all, as a reader, by the conclusions drawn by the AAs. Leukopenia per se (but again, how low should be low the WBCs count) is a well known marker of negative outcome in sepsis / infection and attention has to be drawn in every case of leukopenic pts, whose low/very low WBCs count should start an aggressive differential diagnosis with appropriate tools (PCT / PCR etc) and an appropriate , if considered wise or if indicated, antiinfective broad spectrum therapy .

• Totally agree with deeper investigations to find association(s) between LP and TCP

Reviewer #2: The hypothesis propelling Belok et al. in this manuscript is that patients with leukopenia demonstrate failure of an organ system that is not adequately represented in the 2016 Sepsis-3 definition, which includes the utilization of the SOFA scoring system for new onset organ failure. Their results distinctly show that patients with infection are more likely to die if they present with leukopenia than with a normal or high leukocyte count. Unfortunately, due to the low frequency of leukopenia as a presenting symptom, and is likely an indication of severe disease, it can be attenuated by other markers of organ dysfunction, in this case thrombocytopenia. Nevertheless, it is undoubtedly important for clinicians to recognize leukopenia as a poor indicator of severity or poor prognosis in patients with suspected or confirmed sepsis.

Limitations:

1. The authors indicate that leukopenia could be used as a marker of poor prognosis in patients with sepsis, being that it is a rare occurrence but is associated with increased mortality. In addition, the authors suggest that leukopenia could help identify patients with sepsis who were not identified with Sepsis-3. It would have been interesting for the authors to have discussed whether or not all patients with leukopenia had a diagnosis of sepsis in their charts, or – give that this cohort was from before 2016, to have assessed whether or not they would have all met sepsis criteria.

2. In table 1, the authors should indicate the mean and range for WBC counts for each group; Leukocytosis, normal and leukopenia.

3. Sepsis induced immunosuppression is a key alteration in patients with severe disease and is highly correlated with poor outcomes and mortality. Furthermore, lymphopenia has been shown to correlate with poor outcomes. In the Leukopenia cohort, it would be helpful to know what percentage of patients presented with neutropenia or lymphopenia and how that correlates with mortality. Additionally, it is possible that patients with normal or high WBC counts have lymphopenia as well, and whether or not Leukopenia alone predicts mortality when accounting for differential cell types.

4. The conclusions mention that “leukopenia could be considered a sepsis-defining organ dysfunction”. This statement appears misleading, as the multivariate analysis compared mortality rate with leukopenia to mortality rate without leukopenia in association with other organ systems on the SOFA score, but do not compare mortality rate of the other organ systems at all WBC counts with leukopenia alone to determine that sepsis induced mortality is specifically when associated with leukopenia.

5. Despite the omission of hematologic or hematopoietic failure from the SOFA scoring system, the applicability of these findings is unclear in the manuscript. Even with the absence of leukopenia in diagnosing sepsis, presumably Sepsis-3 is inclusive enough to capture all patients with leukopenia for initiation of sepsis goal directed therapy, and severity of illness will be appreciated based on the current SOFA metric.

6. PLOS authors have the option to publish the peer review history of their article (what does this mean?). If published, this will include your full peer review and any attached files.

Reviewer #1: **Yes: **andrea de gasperi, MD

Reviewer #2: No

---

## [Author Response · Author response to Decision Letter 0]

13 Mar 2021

Dear Editors,

We appreciate the opportunity to submit a manuscript revised to reflect reviewer input and comments. Please find a point-by-point response to your comments and suggests below. 

Regards,

Samuel Belok

 Reviewer #1: 

Aim of this retrospective study in the ICU setting is to try to include leukopenia (WBCs< 4000 / uL , no mention of neutropenia) among the organ dysfunctions in the SEPSIS 3 definition. . “Patients with suspected infection,rather than a diagnosis of sepsis, were evaluated in order to explore the added clinical utility and prognostic validity of leukopenia as a sepsis-defining organ dysfunction – that is, identifying patients with “sepsis” who may not have been identified with the traditional Sepsis-3 definition” .LP per se has long been associated with poor outcome in case of infection. According to the AAs “ identifying leukopenia as an acute organ dysfunction, (rather than as a SIRS criteria), could have ramifications for early identification and expedited treatment of patients with time-sensitive sepsis”.

Leukopenia (LP) ”, the absolute reduced number of white blood cells in the peripheral blood: LP is almost always due to neutropenia or lymphopenia.The terms granulocytopenia and neutropenia are often used interchangeably. The definition of “leukopenia” varies, but in most laboratories the lower limit of a normal total white cell count is 3000/μl to 4000/μl. Neutropenia is defined as an absolute neutrophil count (ANC) of less than 1500/μl, an absolute count less than 800/μl being considered to be associated with an immunocompromised condition. New or severe leukopenia, especially neutropenia, mandates in any case an aggressive management , including a a thorough investigation to determine the cause / source of the leukopenia , to rule out infection / sepsis and an aggressive management (broad short term antiinfective if wise)

Please consider these notes I dare propose to the study

Query 1. LP is since long a marker of severity and a well known risk factor for mortality and morbidity, particularly when severe and in case of absolute neutropenia < 800/ uL. I have not clear the advantage / amelioration of considering LP aa a marker of organ dysfunction (Bone Marrow[BM], I suppose), due to the above reasons . It should be considered per se!

Response1. We set out to evaluate whether leukopenia – as defined by the SIRS criteria and previously included in the Consensus Sepsis Definition Criteria – could contribute to prognostic validity by functioning as a potential marker of organ dysfunction within the current definition of sepsis (increased SOFA score). Initially we did not further explore the driver of leukopenia (e.g. neutropenia vs lymphopenia) because individual white cell lines have not been included in the SIRS criteria. However, we agree that exploring the primary driver of leukopenia (e.g., lymphopenia or neutropenia) in our cohort can aid in the interpretation of results. Therefore we further evaluated the complete blood count differential results for patients in our study cohort. We identified that 5310 (90%) patients within the cohort had complete blood counts with differentials. Of these patients, 167 (3.1%) had neutropenia (<1700 /μL), and 2642 (49.8%) patients had lymphopenia. While the presence of lymphopenia did not add prognostic information, in our exploratory analysis we found that neutropenia did provide additional prognostic information even after adjusting for co-morbidities and components of the SOFA score. 

Manuscript changes by section: 

Methods: “Additionally, we repeated our analyses by substituting neutropenia (defined as neutrophil count less than 1700/μL) and lymphopenia (defined as lymphocyte count less than 900/μL) for leukopenia among patients with a complete blood count differential, in order to assess if the association between WBC and mortality are driven by specific WBC components. We also assessed correlation between neutrophil count and total WBCs and between lymphocyte count and total WBCs using Pearson’s correlation. In our exploratory analyses Pearson’s correlation coefficients were used based on the linearity of the association. 

Results “As an additional exploratory analysis, we evaluated the presence of neutropenia and lymphopenia –rather than leukopenia - among 5310 (90%) patients with complete blood count differentials. Of these patients, 167 (3.1%) had neutropenia (<1700 /μL), 1563 had a normal neutrophil levels (1700 /μL – 7000 /μL), and 3580 (67.4%) had neutrocytosis (>7000 /μL). The median neutrophil count was 9000 /μL (IQR 6000 -1310). In addition, 2642 (49.8%) patients had lymphopenia (< 900 /μL), 2393 (45.1%) had a normal lymphocyte level (900 -1700 /μL), and 275 (5.1)% patients had a lymphocytosis >1700 /μL. The median lymphocyte count was 900 /μL (IQR 500-1500).

 We found that in unadjusted analysis, neutropenia was associated with increased in-hospital mortality compared with neutrocytosis (OR 1.9; 95% CI 1.4 – 2.7). After adjusting for covariates from the Sepsis-3 baseline model (i.e., age, race, gender and comorbities), leukopenia remained associated with increased risk of mortality compared with leukocytosis (OR 2.3, 95% CI 1.6-3.2). After further adjusting for the SOFA score covariates neutropenia remained associated with increased risk of mortality (OR 1.6 95% CI 1.1- 2.3) (Supplemental Table 1). Lymphopenia on the other hand was not associated with increased in-hospital mortality in any of the unadjusted or adjusted models (Supplemental Table 2). Pearson’s correlation coefficient between total WBC count and neutrophils was 0.92 (95% CI 0.92-0.93, p=<0.001) and between WBC and lymphocytes was 0.25 (95% CI 0.22-0.27, p=<0.001).” 

Discussion “However, results from our exploratory analysis showed that neutropenia – but not lymphopenia - was strongly associated with mortality after adjusting for the SOFA scores, suggesting that decreased leukocyte production is a less likely mechanism of the association between leukopenia and mortality during sepsis.”

Q2. The number of pts with LP is in general, and in this series in particular (average ICU pts), small : over 5,909 ICU patients with suspected infection, 250 (4.2%) had leukopenia . It is not specified a WBCs count “stratification” ( < 2000 / < 1000) and how many had severe LP: please, could you provide this figure?

R2. I would like you to direct you to our spline figure (Figure 2) which overlays WBC count with hospital mortality. This shows that the association between leukopenia with mortality is strongest at the most severely leukopenic levels. 

Q3. SOFA median score was in this series 8.5 (mortality 15-20%). WBCs, not present in SOFA, are considered in SAPSII and APACHE II , and are relevant in case of count > 20000/ uL. Close to SOFA 8.5 score are SAPSII 38, , with a mortality of 21.35 ( checked with WBCs 3000 and 1000) and APACHE2 14 (mortality 15%, checked with WBCs count 3000 and 1000: source MEDCALC). According to the mortality prediction curves , minimal seems to be the impact of LP on the score and on mortality

R3. Thank you for the feedback. Our choice of evaluating the prognostic significance of leukopenia in the context of the SOFA score was driven by our primary research question – could leukopenia caused by sepsis be a sign of end organ dysfunction that increases mortality risk? That is – could leukopenia be a sepsis defining organ failure? Thus, we did not evaluate APACHE or SAPS scores which not foundational to the consensus criteria for sepsis. Further, predicted mortality from SOFA, SAPS, or APACHE are likely not calibrated to contemporary sepsis outcomes. 

Q4. Among the SOFA components “platelet count ” (thrombocytopenia. TCP) ) represent “dysfunctioning coagulation”, not a dysfunctioning BM, and, as reported by the AAS, “further adjustment for the platelet component of the SOFA attenuated the association between leukopenia and mortality (OR decreased from 1.5 to 1.1)”. (see page 7).

R4. Thrombocytopenia in sepsis is likely multi-factorial: from increased consumption (Russwurm et. al; citation 14) as well as decreased production (Aird et. al; citation 12) as we describe in our discussion. Thus, the correlation between low platelets and low WBCs may also be due to these correlated mechanisms – either increased consumption via NETs or decreased production in the bone marrow. Our data cannot provide the answer, but provide interesting preliminary data to guide further studies of the mechanisms of correlated platelet-leukocyte levels in sepsis. 

Q5. Indeed, the relevance of LP might be stressed (but the absolute WBCs count is important into my opinion to be reported

R5. Thank you. When you refer to “absolute WBC” we believe you are referring to the Absolute Neutrophil Count? We identified that 5310 (90%) patients within the cohort had complete blood counts with differentials. Of these patients, 167 (3.1%) had neutropenia (<1700 /μL), and 2642 (49.8%) patients had lymphopenia. As detailed in Response 1 above, this analysis has been added to our manuscript.

Q6. according to the AAS, “ 83 (1.4%) of patients had LP without thrombocytopenia and 14 had LP prior to thrombocytopenia”: then LP should be considered an important early warning per se , whether or not considered as the result of a dysfunctioning organ (BM) , and as an early warning and with sepsis / infection suspicion, to be accordingly managed (optimal clinical practice, with further , deeper efforts aiming at a differential diagnosis with appropriate investigations and appropriate antiinfective therapy if wise ) .

R6. We agree that leukopenia may have a role as an early warning sign in sepsis. In the discussion we clarify this point: “Because leukopenia has similar prognostic validity to thrombocytopenia during suspected infection, and is present in some patients prior to onset of thrombocytopenia, leukopenia is potentially a clinically important marker of life threatening hematological dysfunction in patients with suspected infection.”

Q7. The AAs reported a correlation between LP and TCP , able to reduce the prognostic validity of SOFA (OR 1.5 to 1.1); LP , possibly representing an earlier sign of infection / sepsis if compared to TCP, as above quoted, deserves indeed attention, not only in this small subset ofLP patients, but in general (personal opinion, but always applied in clinical practice )

R7. Thank you. I would like to refer you to Table 2 in our chart. The correlation between platelets and leukocytes does not reduce the prognostic validity of SOFA. Rather, the signal of increased mortality associated with leukopenia becomes attenuated when controlling for thrombocytopenia. Because platelet count is part of the SOFA score already, and leukocyte count is correlated with platelet count, the increased mortality seen with leukopenia in analyses unadjusted for the SOFA score is attenuated after incorporating SOFA score. 

Q8. The impact of LP on outcome, as already underlined, might be relevant : much more relevant, however, should be the absolute WBCs count (below 4000, vs below 3000 vs below 1000): one size could not fit all, and the different absolute count might have a different impact on outcome; stratification of WBCs is, into my opinion, mandatory

R8. We agree that degree of leukopenia likely effects mortality as you suggest. Graphically this is represented in Figure 2 where mortality increases as degree of leukopenia worsens. We also performed an exploratory analysis where we analyzed mortality based on absolute neutrophil count and absolute lymphocyte count as described above.

Q9. A way to reinforce the relevance of LP could be the presence of neutropenia instead of “LP” (more specific perhaps in the ICU pts (see Quentin G, Azoulay et al. Influence of neutropenia on mortality of critically ill cancer patients: results of a meta-analysis on individual data , Crit Care, 2018; 22: 326. 2018) : neutropenia ( neutrophil< 1000 u/L) was associated with poor outcome in cancer pts.

I dare suggest the AAS to consider in their study the level of severity of LP and , instead of LP or together with LP , neutropenia: perhaps it might have a greater impact on outcome, particularly in case of severe neutropenia (< 1000/uL or <500)

R9. We have now added an exploratory analysis where we constructed a model using neutropenia and lymphopenia as separate covariates. Additionally, evaluating leukopenia caused by cancer and chemotherapy is not a goal of our study and was an exclusion criteria. We felt that including leukopenia due to malignancy/chemotherapy is out of the scope of exploring definitional criteria from sepsis. While we can report the components of the white count and associations with death to inform some exploratory hypotheses of mechanisms, our primary study question involved the historical characterization of leukocyte as a SIRS criteria. 

Q10. Statistical Methods

Page 5 – A comment / explanation of the statistical methods could be wise, since I think this could be key of the above proposed aim(s) of this study and might shed light on it.

R11. We have added some changes in order to clarify the Statistical methods 

 “Baseline characteristics and values were reported as means +/- standard deviations or percentages, as appropriate. We fitted natural splines to visualize the relationship between mortality rate and WBC count. Nested logistic regression models were used to determine changes to the association between WBC (with leukocytosis as the reference group) and mortality as additional potential confounding variables were added to the analysis and to reflect the processes of model development for the Sepsis-3 definition. We analyzed 1) a univariable model with WBC category alone; 2) a multivariable model that included WBC category and covariates from the Sepsis-3 baseline model; 3) a multivariable model that included WBC category, Sepsis-3 baseline model components, and the SOFA score; and 4) a model including WBC category and SOFA score. We evaluated the odds ratio and the change in the effect estimate for hospital mortality associated with leukopenia as each components and as SOFA score components were added to the model.”

Results

Page 6 – “We identified 5,909 ICU patients with suspected infection. The average age of the cohort was 68.3±16.7 years old, with a mean SOFA score of 8.5±3.69. 3027 patients (51.2%) had leukocytosis and 250 (4.2%) had leukopenia. 1,081 (18.3%) patients died in the hospital1,081 (18.3%) patients died in the hospital.”

Q12. I dare pose two questions :

 How many of the 250 leukopenic pts died in Hospital? In the results we added the mortality of this group as well as the normal WBC group and leukocytosis group. Leukopenia: 68 patients which is 27.2% mortality rate; Normal WBC: 394 patients (15.0%) and leukocytosis group 316 patients (20.4%). We subsequently added this data to table 1. 

How many leukopenic pts were not identified by Sepsis3 Criteria and died because of sepsis ?In-hospital mortality of the entire cohort was 1,081 (18.3%) 

Any difference in mortality between leukopenic pts with absolute WBCs count below 3000 u/L and below 1000 u/L? 

R12. Yes this is visually represented on our spline figure (figure 2). Please also see the answer to Question 6 above for additional responses. 

Q13. Discussion

Page 7 –

• if LP is strongly attenuated by TCP, what is the relevance of its use in the score? The presence of LP has to be taken in any case into account as an early, relevant warning : perhaps the absolute number might impact more than the “absolute” definition . I dare ask a comment

R13. We agree that this was an important finding in our study. We sought to evaluate whether leukopenia added prognostic information to the current framework of the sepsis-3 definition. It did not, due to the correlation between neutrophil count and platelet count, a novel finding. However, these findings do not mean that leukopenia is not a clinically important finding that requires further evaluation. We have changed the manuscript to clarify that lack of contribution of leukopenia to the sepsis consensus definition framework does not mean that leukopenia does not have clinical importance. 

Manuscript clarification:

Conclusion: “Although leukopenia did not independently add to prognostic validity of the SOFA score above platelet count, the correlation of leukopenia with thrombocytopenia and association of neutropenia with mortality suggest that future studies should whether leukopenia or neutropenia may prospectively identify a larger group of patients earlier than current Sepsis-3 definitions.”

Q14. Page 8 -

• Dealing with granulocytopenia (Leibovici et al ,1995), see also Quentin et al, 2018, above quoted , for a short comment : any idea of granulocytopenia and its impact on mortality in this series ? 

R14. Granulocytes is a term often used synonymously with neutrophils in the literature as neutrophils are by far the most common granulocyte in the peripheral blood. 

Q15. I totally agree about the BM dysfunction as a reason for TCP (any reason, as appropriately pointed out by the AAS in the draft), which could share the same cause of LP : but in SOFA the TCP is “dysfunctioning coagulation” (perhaps more appropriate “defective haemostasis”!) and not BM.

R15. Correct, this is how platelets are characterized in SOFA. However, the term “coagulation dysfunction” from SOFA merely describes an observation of low platelets, and despite its implication, does not provide any insights into the mechanism of low platelets – either lack of production or overactive destruction. We discuss how our findings of correlated platelets and WBCs may inform future studies of this mechanism of ‘coagulation dysfunction’ i.e. low platelets in sepsis. 

Q16. Page 9 –

Conclusion

I am sorry to say I am not convinced at all, as a reader, by the conclusions drawn by the AAs. Leukopenia per se (but again, how low should be low the WBCs count) is a well known marker of negative outcome in sepsis / infection and attention has to be drawn in every case of leukopenic pts, whose low/very low WBCs count should start an aggressive differential diagnosis with appropriate tools (PCT / PCR etc) and an appropriate , if considered wise or if indicated, antiinfective broad spectrum therapy .

R16. We agree that leukopenia remains a clinically important finding, even if it does not appear to contribute to the framework of Sepsis-3. Please also refer back to our answer for question 6. We also clarify in the discussion that WBC could be an early warning sign for sepsis: 

“Because leukopenia has similar prognostic validity to thrombocytopenia during suspected infection, and is present in some patients prior to onset of thrombocytopenia, leukopenia is potentially a clinically important marker of life threatening hematological dysfunction in patients with suspected infection.”

Reviewer #2: 

The hypothesis propelling Belok et al. in this manuscript is that patients with leukopenia demonstrate failure of an organ system that is not adequately represented in the 2016 Sepsis-3 definition, which includes the utilization of the SOFA scoring system for new onset organ failure. Their results distinctly show that patients with infection are more likely to die if they present with leukopenia than with a normal or high leukocyte count. Unfortunately, due to the low frequency of leukopenia as a presenting symptom, and is likely an indication of severe disease, it can be attenuated by other markers of organ dysfunction, in this case thrombocytopenia. Nevertheless, it is undoubtedly important for clinicians to recognize leukopenia as a poor indicator of severity or poor prognosis in patients with suspected or confirmed sepsis.

Limitations:

Q17. The authors indicate that leukopenia could be used as a marker of poor prognosis in patients with sepsis, being that it is a rare occurrence but is associated with increased mortality. In addition, the authors suggest that leukopenia could help identify patients with sepsis who were not identified with Sepsis-3. It would have been interesting for the authors to have discussed whether or not all patients with leukopenia had a diagnosis of sepsis in their charts, or – give that this cohort was from before 2016, to have assessed whether or not they would have all met sepsis criteria.

R17. Thank you for the response regarding whether patients with leukopenia would meet prior Sepsis consensus criteria (i.e., Sepsis-1). Since all leukopenic patients had infection and leukopenia, this cohort would require just one more SIRS criterion to be present (fever, elevated respiratory rate or heart rate). This would be interesting to study for future studies - nonetheless we felt that comparing predictive validity of SIRS to Sepsis-3 is out of the scope of this paper. 

Q18. In table 1, the authors should indicate the mean and range for WBC counts for each group; Leukocytosis, normal and leukopenia.

R18. We agree. We included distritution (the median and IQR) of WBC for each group into table 1. Leukopenia: 2.7 (2.0 -3.4); Normal: 8.2 (6.5 – 9.6) and Leukocytosis 15.2 (12.8-19.4)

Q19. Sepsis induced immunosuppression is a key alteration in patients with severe disease and is highly correlated with poor outcomes and mortality. Furthermore, lymphopenia has been shown to correlate with poor outcomes. In the Leukopenia cohort, it would be helpful to know what percentage of patients presented with neutropenia or lymphopenia and how that correlates with mortality. Additionally, it is possible that patients with normal or high WBC counts have lymphopenia as well, and whether or not Leukopenia alone predicts mortality when accounting for differential cell types.

R19. This is valuable feedback. Initially we did not further explore the driver of leukopenia (e.g. neutropenia vs lymphopenia) because individual white cell lines have not been included in the prior SIRS criteria. However, we agree that exploring the primary driver of leukopenia in our cohort can aid in the interpretation of results. Therefore, we further evaluated the complete blood count differential results for patients in our study cohort. We identified that 5310 (90%) patients within the cohort had complete blood counts with differentials. Of these patients, 167 (3.1%) had neutropenia (<1700 /μL), and 2642 (49.8%) patients had lymphopenia. While the presence of lymphopenia did not add prognostic information, in our exploratory analysis we found that neutropenia did provide additional prognostic information even after adjusting for co-morbidities and components of the SOFA score. 

Q20. The conclusions mention that “leukopenia could be considered a sepsis-defining organ dysfunction”. This statement appears misleading, as the multivariate analysis compared mortality rate with leukopenia to mortality rate without leukopenia in association with other organ systems on the SOFA score, but do not compare mortality rate of the other organ systems at all WBC counts with leukopenia alone to determine that sepsis induced mortality is specifically when associated with leukopenia.

R20. We understand. We edited the conclusion statement to say:

“Among critically ill patients with suspected infection, leukopenia was rare, but associated with increased risk of death as compared with leukocytosis. However, the “coagulation” component of the SOFA score (as represented by the platelet count) accounted for the increased mortality risk associated with leukopenia, the presence of leukopenia did not appear to add prognostic information to the current Sepsis-3 criteria.”

Q21. Despite the omission of hematologic or hematopoietic failure from the SOFA scoring system, the applicability of these findings is unclear in the manuscript. Even with the absence of leukopenia in diagnosing sepsis, presumably Sepsis-3 is inclusive enough to capture all patients with leukopenia for initiation of sepsis goal directed therapy, and severity of illness will be appreciated based on the current SOFA metric.

R21. We agree that Sepsis-3 (and specifically the platelet component) is inclusive enough to capture almost all patients with leukopenia. Nonetheless we observed a small subset (~1.6%) in whom leukopenia appeared without or earlier than thrombocytopenia, which may be valuable as an early warning sign of sepsis to a clinician.

---

## [Decision Letter · Decision Letter 1]

12 May 2021

Evaluation of Leukopenia During Sepsis as a Marker of Sepsis-Defining Organ Dysfunction

PONE-D-20-27470R1

Dear Dr. Belok,

We’re pleased to inform you that your manuscript has been judged scientifically suitable for publication and will be formally accepted for publication once it meets all outstanding technical requirements.

Kind regards,

Scott Brakenridge, M.D.

Academic Editor

PLOS ONE

Additional Editor Comments (optional):

Reviewers' comments:

Reviewer's Responses to Questions

**Comments to the Author**

1. If the authors have adequately addressed your comments raised in a previous round of review and you feel that this manuscript is now acceptable for publication, you may indicate that here to bypass the “Comments to the Author” section, enter your conflict of interest statement in the “Confidential to Editor” section, and submit your "Accept" recommendation.

Reviewer #1: All comments have been addressed

Reviewer #2: All comments have been addressed

2. Is the manuscript technically sound, and do the data support the conclusions?

Reviewer #1: Partly

Reviewer #2: Yes

3. Has the statistical analysis been performed appropriately and rigorously? 

Reviewer #1: Yes

Reviewer #2: Yes

4. Have the authors made all data underlying the findings in their manuscript fully available?

Reviewer #1: Yes

Reviewer #2: Yes

5. Is the manuscript presented in an intelligible fashion and written in standard English?

Reviewer #1: Yes

Reviewer #2: Yes

6. Review Comments to the Author

Reviewer #1: thanks for having considered the notes i sent in the new version, able indeed to clarify some aspects of the item you discussed. Main weakness of the work is still present (and addressed ) in the conclusion

Reviewer #2: Thank you for appropriately addressing all of the reviewer comments. The manuscript in its current form is well organized and presents very interesting data in a retrospective cohort study that will undoubtedly spark several future studies.

7. PLOS authors have the option to publish the peer review history of their article (what does this mean?). If published, this will include your full peer review and any attached files.

Reviewer #1: **Yes: **andrea de gasperi

Reviewer #2: No

---

## [Editor Report · Acceptance letter]

9 Jun 2021

PONE-D-20-27470R1 

Evaluation of Leukopenia During Sepsis as a Marker of Sepsis-Defining Organ Dysfunction 

Dear Dr. Belok:

I'm pleased to inform you that your manuscript has been deemed suitable for publication in PLOS ONE. Congratulations! Your manuscript is now with our production department. 

Kind regards, 

on behalf of

Dr. Scott Brakenridge 

Academic Editor

PLOS ONE